# Shifting from Ammonium to Phosphonium Salts: A Promising Strategy to Develop Next-Generation Weapons against Biofilms

**DOI:** 10.3390/pharmaceutics16010080

**Published:** 2024-01-05

**Authors:** Silvana Alfei

**Affiliations:** Department of Pharmacy, University of Genoa, Viale Cembrano, 4, 16148 Genova, Italy; alfei@difar.unige.it

**Keywords:** multidrug resistant (MDR) microorganisms, biofilm, biofilm life cycle, anti-biofilm agents, cationic antimicrobial agents, quaternary ammonium salts (QASs), quaternary phosphonium salts (QPSs)

## Abstract

Since they are difficult and sometimes impossible to treat, infections sustained by multidrug-resistant (MDR) pathogens, emerging especially in nosocomial environments, are an increasing global public health concern, translating into high mortality and healthcare costs. In addition to having acquired intrinsic abilities to resist available antibiotic treatments, MDR bacteria can transmit genetic material encoding for resistance to non-mutated bacteria, thus strongly decreasing the number of available effective antibiotics. Moreover, several pathogens develop resistance by forming biofilms (BFs), a safe and antibiotic-resistant home for microorganisms. BFs are made of well-organized bacterial communities, encased and protected in a self-produced extracellular polymeric matrix, which impedes antibiotics’ ability to reach bacteria, thus causing them to lose efficacy. By adhering to living or abiotic surfaces in healthcare settings, especially in intensive care units where immunocompromised older patients with several comorbidities are hospitalized BFs cause the onset of difficult-to-eradicate infections. In this context, recent studies have demonstrated that quaternary ammonium compounds (QACs), acting as membrane disruptors and initially with a low tendency to develop resistance, have demonstrated anti-BF potentialities. However, a paucity of innovation in this space has driven the emergence of QAC resistance. More recently, quaternary phosphonium salts (QPSs), including tri-phenyl alkyl phosphonium derivatives, achievable by easy one-step reactions and well known as intermediates of the Wittig reaction, have shown promising anti-BF effects in vitro. Here, after an overview of pathogen resistance, BFs, and QACs, we have reviewed the QPSs developed and assayed to this end, so far. Finally, the synthetic strategies used to prepare QPSs have also been provided and discussed to spur the synthesis of novel compounds of this class. We think that the extension of the knowledge about these materials by this review could be a successful approach to finding effective weapons for treating chronic infections and device-associated diseases sustained by BF-producing MDR bacteria.

## 1. Antimicrobial Resistance

Antibiotics have been an essential tool of modern medicine to fight illnesses, but antibiotic treatment is becoming increasingly unreliable as antimicrobial resistance (AMR) has evolved in many bacterial pathogens [1]. Antimicrobial resistance happens when germs like bacteria, fungi, and viruses develop the ability to defeat the drugs designed to kill them [2]. Despite the use of antibiotics initially functioning, germs that have developed resistance continue to grow, making infections increasingly difficult, and sometimes impossible, to treat [3]. Antimicrobial resistance is a naturally occurring process promoted mainly by the misuse and overuse of antimicrobials, thus causing the need for costly new treatments to be developed [4]. Antimicrobial-resistant organisms continue to take thousands of lives per year, including a wide variety of people, such as those within the medical community, public health officials, food producers, pharmaceutical developers, and the general public. Antimicrobial resistance is an urgent global public health threat, with the forecast of 10 million deaths per year globally by 2050 [4]. Gram-negative bacteria, such as *Klebsiella pneumoniae*, *Acinetobacter baumannii*, *Pseudomonas aeruginosa*, *Burkholderia cepacian*, and *Escherichia coli*, are responsible for severe infections including pneumonia, bloodstream infections, wound or surgical site infections, and meningitis in healthcare settings and especially in intensive care units where immunocompromised older patients with several comorbidities are hospitalized [5]. Since antibiotics from the carbapenem family are now the last line of defense against antibiotic-resistant bacterial infections, carbapenem-resistant *Enterobacteriaceae*, extensively drug-resistant (XDR) *P. aeruginosa* and XDR *A. baumannii* are the most clinically relevant resistance phenotypes [5].

Antibiotic degradation, antibiotic target modification, modulation of membrane permeability, structural modifications of bacterial lipopolysaccharide, hyper-expression of efflux pumps, as well as the formation of biofilms (BFs) are some of the recognized mechanisms used by bacteria to develop resistance [6]. Appendix A reports some of the most clinically relevant MDR pathogens with their associated infections [7,8,9,10,11,12,13,14,15,16,17,18,19,20,21,22,23,24,25,26,27,28,29,30,31,32,33,34,35,36,37,38].

To ameliorate this worrying scenario, where antibacterial resistance is incessantly increasing, the available effective antibiotics are decreasing dramatically, and infections sustained by BF-producing pathogens are practically untreatable, researchers have focused mainly on developing drugs acting on contact as membrane disruptors that therefore have a low tendency to develop resistance. In this context, quaternary ammonium salts (QASs) have demonstrated promising biocidal effects and anti-BF potentialities [39,40,41,42,43,44]. Unfortunately, their continuous use over more than 80 years has driven the emergence of QAS resistance [45]. In this regard, quaternary phosphonium salts (QPSs), including tri-phenyl alkyl phosphonium derivatives, achievable by easy, low-cost, one-step reactions and already known as intermediates of the Wittig reaction, have shown promising biocidal and anti-BF effects in vitro. In this paper, to draw more attention to QPSs, which are still too little studied by researchers, after having provided essential information on pathogen resistance and BFs and a discussion on QASs, we have reviewed the QPSs developed and assayed to this end in recent years. Finally, to spur the synthesis of novel compounds of this class, potentially active against BFs, the synthetic strategies used to prepare the antibacterial QPSs developed so far have been also provided and discussed.

## 2. Biofilms: The Antibiotic-Resistant Home Self-Produced by Microorganisms

A particularly worrying form of pathogen resistance is represented by biofilms (BFs), which are a safe and antibiotic-resistant home to microorganisms [46]. BFs are well-organized heterogeneous communities of self-immobilized microorganisms, attached to each other and to a surface, which are encased within self-produced extracellular polymeric substances (EPSs). EPSs protect them from environmental stressors, including extreme conditions of pH, oxygen, osmotic shock, heat, freezing, UV radiation, predators, disinfectants, and antibiotics [47]. In BFs, cell–cell interactions are facilitated, including those that allow the transfer of resistance genes, and microorganisms cooperate in a synchronized way to capture resources, such as nutrients [7,48]. BFs are a lifestyle for 40–80% of prokaryotic or unicellular eukaryotic cells [49], representing an incubator for genotypic and phenotypic variants [48]. In fact, BFs should be regarded as a multicellular superorganism where sessile cells, persistent cells, and dormant cells, with physiological characteristics differentiating them from planktonic cells, cohabit and communicate by the quorum sensing (QS) system [7,48]. The extracellular polymeric matrix in which BF-producing cells are encased consists of extracellular polymeric substances (EPSs), including a combination of enzymatic proteins, polysaccharides (cellulose, poly-glucosamine (PGA), and exopolysaccharides), extracellular DNA (eDNA), as well as cationic and anionic glycoproteins and glycolipids, which allow real communication between the bacteria and stabilize the three-dimensional structure of the BF itself [50,51]. Appendix A schematizes the general composition of BF biomass. 

EPSs not only protect bacteria from external dangers but also provide them with the necessary nourishments and water, which is retained by means of H bonds with hydrophilic polysaccharides. Depending on the required nutrients, the microorganisms in a BF can change the composition of the EPSs by secreting proper signaling molecules [50,51]. Although beneficial microorganisms could form BFs associated with human, animal, and plant hosts as an essential part of the holobiont [52,53], BFs produced by pathogens are responsible for about 80% of bacterial infections [54], which are often extremely difficult to treat due to the specific protection mechanisms provided by the BF to bacterial cells, immobilized within the EPSs and not reachable by antibiotics [55]. Indeed, BF-producing cells are 10–1000-fold less susceptible to various antimicrobial agents than their planktonic forms. BF-related infections can be divided into device-associated infections and non-device-associated ones. Table 1 collects the most relevant BF-producing pathogens and their related infections. 

### 2.1. Biofilm Formation and Dispersal

Bacteria can exhibit a free-living (mobile) planktonic state of living and a sessile (non-mobile) biofilm-living state when adhered to a substrate [72]. The in vitro BF life cycle involves the different stages reported in Appendix A [73] and visualized in Figure 1.

At stage 5 concerning the detachment and dispersion of cells from the BF, the initiation of new BF formation has been observed. Dispersed cells are morphologically more like planktonic cells than those of mature BFs and are crucial for developing into a new BF [61]. Reproducible periodic BF dispersal in bacterial species such as *Actinobacillus* spp., *Actinomycetes comitans* [75], *P. aeruginosa* [76], and *Serratia marcescens* [77] has been reported. The main factors, cues, and signals that induce BF detachment include temperature, pH, enzymatic degradation, starvation or excess of nutrients, presence of D-amino acids, nitric oxide, biosurfactants, and QS signaling molecules. In vitro and in vivo experiments have shown that the cells liberated by BFs are extremely cytotoxic to macrophages, more sensitive to iron diminution, and significantly more virulent to nematode hosts than planktonic bacteria. Additionally, it was reported that dispersed bacteria derived from BFs treated with glycoside hydrolase rapidly induced fatal septicemia in a mouse chronic wound infection model.

#### 2.1.1. Quorum Sensing (QS) in Bacterial Biofilms

The formation of BF is governed by physiological processes and by the coordinated activities of the population of germs, which communicate with each other mainly through the quorum sensing (QS) machine [78]. Particularly, QS is a specific intercellular communication method based on the release of signal molecules that diffuse in BFs and are detected by other microorganisms, which in turn respond. QS machinery plays a vital role in BF formation, exopolysaccharide synthesis, motility, and chemotaxis, which are imperative for bacteria during the degradation or detoxification of any pollutant [78]. QS, first reported in the marine bioluminescent bacterium *Vibrio fischeri* [79], is currently the paramount target of strategies to control BF formation by disrupting cell-to-cell communication, conjugation, nutrient acquisition, and even motility and production of certain metabolites [78]. QS signaling molecules are different in Gram-positive and Gram-negative bacteria. There are at least three types of QS systems, including the acyl-homoserine lactone system (AHL) in Gram-negative bacteria, the autoinducing oligopeptide (API) QS system in Gram-positive bacteria, and the LuxS-encoded autoinducer-2 QS system (AI-2) in both species [80]. AHL and API QS systems, which are based on signaling molecules such as acyl-homoserine lactones and small peptides, respectively, are planned specifically for “intraspecies” signaling. An increase in AHL molecules corresponds to high bacterial growth [81], while high concentrations of APIs affect target gene expression, which are responsible for the production of numerous toxins and degradable exoenzymes [78]. The production of AI-2 signals, also called “universal language”, is catalyzed by LuxS synthase and allows communication between microorganisms of different species [82]. Moreover, LuxS is involved in the activation of the methylation cycle, which is involved in the control of the expression of hundreds of genes associated with the microbial processes of surface adhesion, detachment, and toxin production [82]. 

#### 2.1.2. Cyclic Dimeric Guanosine Monophosphate (c-di-GMP) in Bacterial Biofilms

Bis-cyclic dimeric guanosine monophosphate (c-di-GMP) is a secondary messenger that mediates crucial cellular activities in bacteria, including growth, motility, virulence, BF formation, and cell cycle progression [83]. In particular, c-di-GMP is involved in the transformation of planktonic cells to sessile ones during BF formation, as well as in the passage from the sessile lifestyle to the planktonic one through the BF dispersal process [84]. The BF formation and dispersal processes are governed by c-di-GMP through several genetic interactions. In Gram-negative bacteria, low and high c-di-GMP levels correlate with the motile and sessile phenotypes, respectively [83], so high concentrations lead to BF formation, while low concentrations stimulate motility and thus dispersal of bacterial cells from BF [84]. Such a correlation has been demonstrated especially in *E. coli*, *P. aeruginosa*, and *Salmonella typhimurium* [85].

## 3. Current and New Therapeutic Approaches against BF

Infections caused by BFs are a worrying problem causing chronic diseases that are difficult to treat and require high-dose antibiotics according to the severity of infection. As previously reported, cells in BFs are up to 1000-fold more tolerant to antibiotics and disinfectants than planktonic ones [86]. Additionally, when infections are device-associated and antibiotics fail due to the increasing resistance of bacteria in BFs, surgical replacement of the infected devices is necessary [87]. In this regard, it has been reported that the best possible treatment for BF-based infections could be the possibility of an early diagnosis using biosensors, advanced imaging, or theranostic nanoparticles (NPs), as well as inhibiting the initial attachment stage, thus preventing the infection from starting [86]. Innovative and effective antibiotic strategies have been designed to counteract BF formation and promote its dispersal, such as treatments for dispersing BFs by dissolution of EPSs, associations of conventional antibiotics with QS inhibitors, and a mixture of these novel techniques. Further advanced treatments include antimicrobial photodynamic therapies, bioacoustics, as well as electric and magnetic fields [88]. However, this research area is still too little explored and far from undergoing clinical research and entering the commercial market [88]. Currently, there are a few molecules under clinical development that are active against MDR pathogens and/or BF producers. Among the molecules in Phase III clinical trials in 2020, only caspofungin, which is an antifungal drug, has been demonstrated to inhibit the synthesis of the polysaccharide components of the *S. aureus* BF [89]. In the field of NPs, polymeric lipid NPs deriving from the conjugation of rhamnolipids (biosurfactants secreted by *P. aeruginosa*) and polymer NPs made of clarithromycin encapsulated in a polymeric core of chitosan have been studied, while rhamnolipid-coated silver and iron oxide NPs have been developed, which were effective in eradicating *S. aureus* and *P. aeruginosa* BFs [89]. Naturally occurring cationic antimicrobial peptides (CAMPs), a class of non-β-lactam antimicrobial agents, acting on contact as membrane disruptors, have been demonstrated to be effective on a wide variety of Gram-positive and Gram-negative bacteria, fungi, protozoa, and yeast and to be active alone or in combination with conventional antibiotics against *Staphylococci* and *P. aeruginosa* BFs [90,91,92,93]. So, inspired by CAMPs, several cationic compounds have been developed that demonstrate similar mechanisms of action and effects. In this context, some antimicrobial cationic dendrimers have proven anti-BF effects and the capability to penetrate EPSs and prevent BF formation, thus representing novel promising substances to counteract this alarming form of bacterial resistance [7,88]. Quaternary ammonium salts (QASs) include the most recognizable commercial disinfectants and antiseptics, such as benzalkonium chloride [39], miramistin [40], di-decyl-dimethyl-ammonium chloride [41], cetyl-pyridinium chloride [42], and octenidine dichloride [43]. Moreover, studies reported that surfactants classified as gemini quaternary ammonium salts (g-QASs) have been inserted in biocidal preparations which were active against the biofilm created by *P. aeruginosa* by disrupting its functioning, and lysing BF cells upon electrostatic interactions [44]. In the case of BFs formed by Gram-positive bacteria, g-QASs such as alanine gemini surfactants manifested anti-BF activity against *S. epidermidis* [44]. With the aim of preventing the phenomenon of adhesion of microorganisms, surfaces modified with QASs and their gemini derivatives have been developed. Moreover, the ability of QASs to eradicate *C. albicans* and *R. mucilaginosa* BFs was observed [44]. 

### 3.1. Nitrogen-Based Quaternary Salts

Permanently cationic nitrogen-based surfactants, such as quaternary ammonium salts (QASs) and derivatives are made of one or more polar cationic heads and differently structured hydrophobic carbon tails, often including long carbon chains. QASs possess algistatic, bacteriostatic, tuberculostatic, sporostatic, fungistatic, and antiviral activities [94,95,96,97]. Although their real mechanism of action has not been fully understood, as schematically shown in Figure 2, it has been reported that at a concentration close to the MIC, after the initial adsorption of QASs on the cell wall of pathogens by electrostatic interaction, they penetrate it, due to the presence of hydrophobic alkyl chains [98]. 

Here, QASs react with lipids and proteins of the cell membrane, thus causing disorganization in its structure, pore formation, loss of osmoregulatory and physiological functions, and leakage of low-molecular-weight components out of the cell [98]. Subsequently, proteins and nucleic acids degrade inside the cell, and the release of autolytic enzymes occurs, which leads to the lysis of the cell wall components and a complete loss of the structural organization of the cell [98]. At concentrations higher than the MIC, aggregates are formed that solubilize hydrophobic membrane elements [99]. It has been shown that the highest biocidal activity against Gram-positive bacteria and yeast was provided by QASs with C10–C14 carbon chains, while QASs with C14–C16 carbon chains were biocidal against Gram-negative bacteria [100]. However, to exert their biocidal effects, QASs need to pass through the outer envelope of the bacterial cells. In this regard, since that of Gram-negative species is composed of more layers than that of Gram-positive strains, it is not surprising that Gram-negative bacteria show more resistance to QASs than Gram-positive species [101]. Concerning BFs, the inhibition of microbial adhesion to surfaces seems to be a good target for their prevention, and numerous studies have focused on surface modification in order to impede the interaction of microbial cells with it. In this context, the immobilization of QASs on surfaces has been reported as a successful way of reducing bacterial cell attachment. In the past years, the groups of Tan and Peng have shown that chitosan derivatives of monomeric QASs deposited on bone cement and titanium inhibited the adhesion of *Staphylococci* [102,103], while their incorporation into silica NPs specifically reduced the adhesion of *S. aureus* [104]. Although the anti-adhesion effect of QASs also depends on the material properties of the surfaces (hydrophobicity, charge) and on the characteristics of the cell envelope, it has been demonstrated that the presence of the second QAS molecule in the gemini structures (Figure 3) enhances their surface activity, resulting in a stronger interaction with the material [105].

Concerning this, polystyrene coatings decorated with glycine and alanine gemini QASs reduced the adhesion of *S. epidermidis* and *C. albicans*, with the best effects achieved for C12 and C14 compounds. A greater challenge is represented by eradicating mature BFs due to the presence of EPSs and sessile cells with low metabolic activity. Glycine- and alanine-derived gemini QASs with 12 carbons in the chains exhibited not only high antimicrobial effects against planktonic forms of bacteria and fungi but also impressive biofilm-eradicating properties. Specifically, gemini-like surfactants were able to penetrate the BF structure by extracting adherent cells or lowering the viability of cells within the BF. In 2013, Obłak et al. synthesized a series of cationic gemini chlorides and bromides, which poorly reduced the adhesion of microorganisms to the polystyrene plate. However, one compound eradicated the BFs of *C. albicans* and *Rodotorula mucilaginosa*, thus resulting in promise in overcoming catheter-associated infections [106]. Later, Tran et al. developed polyurethane (PU) foam wound dressings coated with poly diallyl-dimethylammonium chloride (pDADMAC–PU), which was able to inhibit the growth and development of BFs of *S. aureus*, *P. aeruginosa*, and *A. baumannii* within the wound dressing. A colony-forming unit assay revealed that the pDADMAC–PU dressing produced more than an eight-log reduction in the BF formation of each pathogen [107]. More recent examples of the nitrogen-based permanently cationic materials studied as anti-BF agents have been included in Table 2, which collects mainly the foremost compounds developed in the last five years. 

### 3.2. Quaternary Phosphonium Salts (QPS)

Although they are thermally and chemically more stable [154,155,156,157,158,159] and often display increased antimicrobial properties and lower toxicity than their nitrogen-based counterparts [155,160,161,162], phosphonium compounds as antibacterial agents are inexplicably lesser reported than ammonium ones. [Fig pharmaceutics-16-00080-ch001] reports the chemical structures of some QPSs developed in recent years that have shown remarkable antibacterial and anti-biofilm effects.

Nevertheless, compounds with phosphonium moieties have been reported and used in various biomedical applications, such as antibacterials [163], anticancer agents [164], and water treatment [165,166]. Concerning their mechanism of action, like CAMPs and QASs, QPSs also act on contact as membrane disruptors, following the phases reported in Figure 2. However, it has recently been demonstrated that some representatives of this class, including alkyl triphenyl phosphonium and alkyl tributyl phosphonium salts, are also endowed with remarkable antioxidant activity, thus being capable of reducing oxidative stress triggered by bacteria, helping the human body in fighting them and blocking the replication of microbial pathogens [158]. In the past, the antioxidant properties of tetrakys (THPS) and its biocidal activity by disrupting the disulfide bonds in bacterial cells were reported by Keasler et al. (2010) [149]. The remarkable antimicrobial effects of phosphonium ionic liquids (PILs) have been reported by Brunel et al. [167]. Benzyl triphenyl phosphonium salts exhibited anti-trypanosomal activity against *Trypanosoma brucei*, and it was established that the existence of bulky substituents (alkyl or aryl) surrounding the phosphonium ions resulted in a reduction in the MIC of these ILs [168]. Bis-phosphonium salt-derived benzophenone exhibited remarkable toxicity to the human protozoan parasite Leishmania [169]. More recently, Das et al. reported the development of mono- and bis-PILs through a purely ionic approach, and the resulting PILs demonstrated selective bacterial toxicity toward Gram-positive *S. aureus* and Gram-negative *E. coli* depending on the number of phosphonium ions present in the salt [154]. Phosphonium salts with single and double alkyl chains were investigated by Endo and coworkers [160]. Herein, the effect of the chain length (C10–C18) on antibacterial activity against a wide range of pathogens including MRSA was examined. The study brought forward a direct correlation between molecular structure and antibacterial activity. In particular, it was observed that higher alkyl chain lengths and the presence of aryl groups improved the antimicrobial activity of PILs. However, the aforementioned studies reported the specific use of phosphonium salts as biocides, while studies conducted on their potential activity against BFs are very limited. In this regard, Kim et al. reported on the antifouling properties of tributyl tetradecyl phosphonium chlorides in reverse osmosis processes, which were capable of reducing the thickness and volume of a BF of *P. aeruginosa* [170]. Among QPSs, triphenyl alkyl phosphonium compounds, known for years as chemical intermediates of the Wittig reaction, easily synthesizable by a one-step low-cost reaction and easily purifiable by crystallizations, have been re-evaluated as promising template molecules to prepare new antibacterial agents active against bacterial BFs. In this regard, a report by Fernández and co-workers described several bioeffects of numerous alkyl-triphenyl phosphonium salts, including the broad-spectrum biocidal activity of compounds with alkyl chains > C7, the antifouling properties of some of them, and the abilities of some others to act as non-biocidal, non-toxic QS disruptors, thus without tendency to develop resistance in microorganisms [171]. In this regard, Melander and co-workers pointed out that it would be extremely important and desirable to develop anti-biofilm agents operating via non-biocidal mechanisms for several reasons, the most important one being avoiding resistance development [172]. In this context, Joseph et al. found that while ammonium and methyl imidazolium cationic pillar[n]arenes were effective inhibitors of BF formation in several strains of Gram-positive bacteria, they were deprived of any antimicrobial activity, thus not causing damage to red blood cells or toxicity to human cells in culture [152]. In the same year, the same authors prepared both ammonium and phosphonium derivatives, which demonstrated similar anti-biofilm properties [153]. Collectively, to have an idea of the discrepancy that exists between the interest in QASs rather than QPSs as possible anti-biofilm agents, Figure 4 shows how many experimental publications have been developed in the last 15 years reporting on quaternary ammonium salts (602) vs. those reporting on quaternary phosphonium salts (13) that were demonstrated to be effective against BFs and BF-based biofouling according to Scopus and PubMed.

Unfortunately, the paucity of innovation evidenced in Figure 4 concerning “onium” compounds has driven the emergence of QAS resistance, and nowadays it is necessary to identify the next generation of disinfectant molecules with efficacy against highly resistant BF-producing clinical isolates [45]. In this regard, a more intensive study on phosphonium-containing compounds could be a rational route to achieve next-generation disinfectants. As confirmation of this thesis, Michaud et al. recently tested a series of quaternary ammonium and quaternary phosphonium compounds (QPCs) against a panel of 35 resistant *A. baumannii* clinical isolates. The authors found that in contrast to QASs, which developed resistance, a QPC, namely P6P-10,10, maintained efficacy against resistant strains with an IC_90_ of 3 µM and an MBEC as low as 32 µM against extensively drug-resistant clinical isolates [45]. Furthermore, this year, Shi et al. reported on the synthesis, characterization, and microbiological evaluation of alkyl-bis-(triphenyl) phosphonium bromides, namely (1,2-DBTPP)Br_2_, (1,4-DBTPP)Br_2_, and (1,6-DBTPP)Br_2_. The biological results revealed that the butyl derivative (1,4-DBTPP)Br_2_ exhibited low toxicity and low hemolytic activity on eukaryotic cells and the capability to inhibit bacterial growth and BF formation and to promote the recovery process of infected wounds in vivo [173]. 

In addition to those discussed here, the QPSs developed and studied in the last 15 years have been included in Table 3.

## 4. Synthetic Strategies Applied to Prepare the Aforementioned QPSs

In this last section, the synthetic procedures followed to synthesize the best-performing QPSs included in Table 3 have been reported, providing reaction Schemes, molecule structures, and some more detail.

### 4.1. Synthesis of Chitosan-Based QPSs

Chitosan QPSs were prepared by means of a two-step reaction. The first step consisted of the synthesis of quaternary benzyl phosphonium salts with a *p*-carboxylic acid group, which was used to graft them onto commercial chitosan, thus obtaining the related chitosan-based QPS derivatives (Figure 1).

Particularly, 4-(chloromethyl) benzoic acid was dissolved in ethyl acetate (EtOAc) and treated with trimethyl phosphine or triphenylphosphine under constant magnetic stirring. The mixture was heated to 85 °C for 10 h, then cooled to room temperature. The precipitate was collected by filtration and subjected to repeated washing with EtOAc to afford the desired products, which were dried in a vacuum at 60 °C for 24 h.

### 4.2. Synthesis of Tetrakis (Hydroxymethyl) Phosphonium Salt (THPS)

A very effective method of preparing tetrakis (hydroxymethyl) phosphonium salt (THPS) consists of reacting phosphine (PH_3_) with excess formaldehyde in the presence of hydrogen chloride, according to Figure 2.

### 4.3. Synthesis of Bis-Phosphonium Salts of Pyridoxine

The bis-phosphonium salts reported by Kayumov et al. (namely compounds **4**, **5**, and **6** in their paper) can be prepared following the procedure described and then optimized by Pugachev et al. [178,179]. Briefly, the chlorine derivatives of pyridoxine **1** (compounds **2** and **3**) were prepared by first introducing the ketal or acetal protections using acetone or butyl aldehyde, respectively, in acidic conditions. Then, an additional hydroxymethyl group was introduced into position 6 of the pyridine ring of pyridoxine by hydroxy methylation in an alkaline medium. Finally, by treating the hydroxyl derivatives with SOCl_2_ at room temperature, **2** and **3** were achieved, which were treated with a 2.0-fold molar excess of triphenyl or tributyl phosphine in CH_3_CN at reflux temperature for 7 h, achieving compounds **4**, **6,** and **5**, respectively (Figure 3).

### 4.4. Synthesis of Pillar [5]Arene-Based QPSs

The pillar [5]arene-based QPSs **1** and **2** and monomer (**M**) reported by Joseph et al. [153] were prepared according to Figure 4.

Briefly, compound **1a** was prepared by refluxing hydroquinone and a strong excess of 1,3-dibromopropane in the presence of potassium carbonate (K_2_CO_3_) in acetone for 24 h under an argon atmosphere. The reaction mixture was then cooled to 25 °C and filtered through celite, the solvent was evaporated under vacuum, and the residue was dissolved in dichloromethane, washed with water, 3 N HCl, and brine, dried with sodium sulphate, and concentrated in vacuo. After purification by column chromatography and recrystallization, the obtained white solid was converted into compound **1b** by dissolution in 1,2-dichloroethane and treatment with paraformaldehyde followed by BF_3_·OEt_2_ at 30 °C for 30 min under an argon atmosphere. Compound **1b** was obtained as a white solid after precipitation in methanol and purification by chromatography. Finally, by refluxing **1b** in a pressure tube in acetonitrile with trimethyl phosphine or triethyl phosphine dissolved in tetrahydrofuran (THF) for 72–96 h, compound **1** or **2** was obtained as white solids in an 86% or 41% yield. Monomer **M** was prepared as compound **1** but starting from compound **1a** rather than **1b**.

### 4.5. Synthesis of Tributyl Tetradecyl Phosphonium Chloride (TTPC)

Tributyl tetradecyl phosphonium chloride (TTPC) was easily synthesized following the procedure proposed by Swapnil Dharaskar and Mika Sillanpaa [180] (Figure 5).

Briefly, tri-butyl-phosphine was added to an equimolar amount of 1-chlorotetradecane at 140 °C under a nitrogen (N_2_) atmosphere and was vigorously stirred for 12–16 h. After reaction completion, the mixture was vacuum stripped to remove volatile organic components and excess 1-chlorotetradecane. TTPC was obtained as a clear, pale-yellow liquid.

### 4.6. General Synthesis of Alkyl Triphenyl Phosphonium Bromide (ATPB)

The most common method to prepare the alkyl triphenyl phosphonium salts used by Martín-Rodríguez et al. in their study [171] involves the quaternization of triphenyl phosphine with the proper alkyl halides at 140 °C, according to Figure 6 [181].

### 4.7. Synthesis of Tetra Alkyl Tetraphenyl Bis-Phosphonium Compound (TATPBP, Namely P6P-10,10)

The synthesis of the best-performing biocidal QPS reported by Michaud et al. [45], was previously reported by the same authors (Figure 7) [182].

Briefly, to 1,6-bis (diphenyl phosphino) hexane (**1**) in acetonitrile, 1-bromodecane was added. The obtained solution was heated to reflux and stirred for 24 h. After cooling to room temperature and solvent removal, an oil was obtained, which was triturated with 1/1 ether/hexane and cooled at −25 °C overnight. The resulting precipitate was dissolved in dichloromethane and then concentrated to afford P6P-10,10 as a white crystalline powder, in a 94% yield.

### 4.8. Synthesis of Alkyl Triphenyl Bis-Phosphonium Bromides

A simple one-step reaction of triphenylphosphine with proper dibromo alkanes was carried out to obtain the alkyl-bis-(triphenyl) phosphonium bromides (1,2-DBTPP)Br_2_, (1,4-DBTPP)Br_2_, and (1,6-DBTPP)Br_2_ (Figure 8).

Briefly, 1,2-, 1,4-, and 1,6-haloalkanes and Ph_3_P were dissolved in DMF and were reacted at 130 °C for 4 h. The mixtures were then cooled to room temperature, and the solvent was removed by rotation volatilization to extract the raw products, which were washed with *n*-hexane and dried in an oven.

### 4.9. Synthesis of N-Phosphonium Chitosans (NPCSs) with Different Degrees of Substitution

*N*-phosphonium chitosans (NPCSs) with different degrees of substitution (3%, 13%, and 21%) were synthesized according to previous studies by the same authors [183,184] (Figure 9).

In brief, chitosan and hydroxybenzotriazole (HOBt) were stirred in an H_2_O/DMSO (*v*/*v* = 2/1) mixture overnight at 15 °C until the homogeneous solution was obtained. Then, commercial CTPB dissolved in H_2_O/DMSO (*v*/*v* = 2/1) was added to the solution, followed by the dropwise addition of a solution of *N*-(3-dimetilaminopropil)-*N*′-etilcarbodiimide hydrochloride (EDC·HCl) in DMSO. After a 24 h reaction at 15 °C, the mixture was precipitated in diethyl ether/acetone (*v*/*v* = 1/2). The product was further purified using a dialysis tube (molecular weight cut-off, MWCO 3500 Da) with distilled water. The final product was obtained by lyophilization. 

## 5. Conclusions

The main scope of this review was to attract more attention to a class of quaternary “onium” salts, such as quaternary phosphonium salts (QPSs), extensively studied for other purposes but still little considered as new antibacterial agents in comparison to their nitrogen-based counterpart. In this context, the study of QPSs as possible next-generation weapons to prevent BF formation or eradicate mature BFs is even more limited. In fact, we have underlined that in the last 15 years, publications concerning the anti-biofilm effects of QPSs total only 13 vs. the 602 concerning quaternary ammonium salts (QASs). However, the paucity of innovation in this space has driven the emergence of QAC resistance by different mechanisms including the overexpression of efflux pumps. On the contrary, there could be several advantages of employing QPSs in place of QASs, including less tendency to develop resistance, low toxicity towards eukaryotic cells, higher antibacterial and biocidal effects, capacity of quorum sensing (QS) disruption, and higher thermal stability. Additionally, the synthetic procedures useful to prepare the most promising antibacterial QPSs developed so far have been reviewed, to stimulate scientists to synthesize new QPS derivatives and enlarge the research concerning these compounds, which are still too little studied, and their antimicrobial effects. We are confident that the extension of knowledge on these materials by this review could be a successful approach to finding effective new weapons for treating chronic infections sustained by biofilm-producing MDR pathogens and environmental issues associated with biofilm-sustained fouling.

## Data Availability

The data presented in this study are available in this article (and Appendix A).

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
