# Peer review of "Shifting from Ammonium to Phosphonium Salts: A Promising Strategy to Develop Next-Generation Weapons against Biofilms"

_pharmaceutics, 2024, doi:10.3390/pharmaceutics16010080_

Round 1
Reviewer 1 Report
Comments and Suggestions for Authors
Shifting From Ammonium to Phosphonium Salts: A Promising Strategy to Develop Efficient Next Generation Weapons Against Biofilm
General
In the title, the abstract, page 20, line 341 and page 28, line 489, the authors suggest that the phosphonium compounds are superior over quaternary ammonium compounds, which is not in line by taking into account the much easier way to prepare QACs and their versatility.
Citation line 341:” However, a paucity of innovation in this space has driven the emergence of QACs resistance”. So, I don’t agree with this statement. This statement is not underlined with facts. Moreover, QACs are widely on the market and used extensively for more than 80 years, in many applications without inducing resistance. It would be astonishing if they can now suddenly induce resistance. Moreover, phosphonium are very interesting compound and there is no need to go in competition with arguments which are not justified. The authors mentioned in the conclusions a lot of advantages (line 491) of quaternary phosphonium compounds over quaternary ammonium compounds, which are not discussed in the paper.
The paper is composed of two parts. The first part is a review on bacteria involved by implants. The second part is on quaternary ammonium and phosphonium biocides that can on kill these bacteria. This combination fits quite well.
It is a comprehensive overview and worthwhile to publish. However, it is exaggerated to state that the quaternary phosphonium compounds are preferred over QACs, as they are much more elaborated to prepare.
A few minor remarks:
Line 42: “….to treat [3]” should end with a point (.).
Table 2: “…Water (up to 97%)” is of course with respect to the “dry” compounds. Maybe better to say: 0-97%.
Page 7. The images of figure 1 are taken from the literature and modified a bit. The reference should be given.
Page 24, line 386: Citatiion”….presence of chloride acid, according to Scheme 2”. Chloride acid should be: hydrogen chloride.
Page 28, line 495: Citation:”….to show researcher how simple and low cost they are…”. The syntheses as described in schemes 3-9 are far from simple and low cost”. I understand that the authors would like to promote the phosphonium compounds and their properties are indeed very good, but please don’t exaggerate to call it cheap. Underline the good properties, as they are present. See also the remark on QACs.

Author Response
General
In the title, the abstract, page 20, line 341 and page 28, line 489, the authors suggest that the phosphonium compounds are superior over quaternary ammonium compounds, which is not in line by taking into account the much easier way to prepare QACs and their versatility.
I thank a lot the Reviewer for his/her comment. Anyway, I would make kindly note to the Reviewer, that what I have reported in my paper has been found in literature. In fact, it is largely documented that QPSs are often better than their nitrogen containing counterparts, in terms of lower cytotoxicity (Errazquin, D., Mohamadou, A., Dupont, L. et al. Ecotoxicity interspecies study of ionic liquids based on phosphonium and ammonium cations. Environ Sci Pollut Res 28, 65374–65384 (2021). https://doi.org/10.1007/s11356-021-14851-0; Shi LW, Zhuang QQ, Wang TQ, Jiang XD, Liu Y, Deng JW, Sun HH, Li Y, Li HH, Liu TB, Liu JZ. Synthetic Antibacterial Quaternary Phosphorus Salts Promote Methicillin-Resistant Staphylococcus aureus-Infected Wound Healing. Int J Nanomedicine. 2023 Mar 7;18:1145-1158. doi: 10.2147/IJN.S398748. PMID: 36915699; PMCID: PMC10007997), higher biodegradability (Atefi, F.; Garcia, M.T.; Singer, R.D.; Scammells, P.J. Phosphonium Ionic Liquids: Design, Synthesis and Evaluation of Biodegradability. Green Chem. 2009, 11, 1595–1604, doi:10.1039/B913057H), higher activity (Michaud, M.E.; Allen, R.A.; Morrison-Lewis, K.R.; Sanchez, C.A.; Minbiole, K.P.C.; Post, S.J.; Wuest, W.M. Quaternary Phosphonium Compound Unveiled as a Potent Disinfectant against Highly Resistant Acinetobacter Baumannii Clinical Iso-lates. ACS Infectious Diseases 2022, 8, 2307–2314, doi:10.1021/acsinfecdis.2c00382; Kanazawa, A.; Ikeda, T.; Endo, T. Synthesis and Antimicrobial Activity of Dimethyl- and Trimethyl-Substituted Phosphonium Salts with Alkyl Chains of Various Lengths. Antimicrobial Agents and Chemotherapy 1994, 38, 945–952, doi:10.1128/aac.38.5.945; Kurata, S.; Hamada, N.; Kanazawa, A.; Endo, T. Study on Antibacterial Dental Resin Using Tri-n-Butyl(4-Vinylbenzyl)Phosphonium Chloride. Dental materials journal 2011, 30 6, 960–966), higher thermal and chemical stability (Cieniecka-Rosłonkiewicz, A.; Pernak, J.; Kubis-Feder, J.; Ramani, A.; Robertson, A.J.; Seddon, K.R. Synthesis, An-ti-Microbial Activities and Anti-Electrostatic Properties of Phosphonium-Based Ionic Liquids. Green Chem. 2005, 7, 855–862, doi:10.1039/B508499G; Das, S.; Paul, A.; Bera, D.; Dey, A.; Roy, A.; Dutta, A.; Ganguly, D. Design, Development and Mechanistic Insights into the Enhanced Antibacterial Activity of Mono and Bis-Phosphonium Fluoresceinate Ionic Liquids. Materials Today Communications 2021, 28, 102672, doi:10.1016/j.mtcomm.2021.102672; Cieniecka-Rosłonkiewicz, A.; Pernak, J.; Kubis-Feder, J.; Ramani, A.; Robertson, A.J.; Seddon, K.R. Synthesis, An-ti-Microbial Activities and Anti-Electrostatic Properties of Phosphonium-Based Ionic Liquids. Green Chem. 2005, 7, 855–862, doi:10.1039/B508499G; Shive, P.N.; Diehl, J.F. Reduction of Hematite to Magnetite under Natural and Laboratory Conditions. Journal of geomagnetism and geoelectricity 1977, 29, 345–354; Metelytsia, L.O.; Hodyna, D.M.; Semenyuta, I.V.; Kovalishyn, V.V.; Rogalsky, S.P.; Derevianko, K.Y.; Brovarets, V.S.; Tetko, I.V. Theoretical and Experimental Studies of Phosphonium Ionic Liquids as Potential Antibacterials of MDR Acinetobacter Baumannii. Antibiotics 2022, 11, doi:10.3390/antibiotics11040491; Hemp, S.T.; Zhang, M.; Allen, M.H.; Cheng, S.; Moore, R.B.; Long, T.E. Comparing Ammonium and Phosphonium Pol-ymerized Ionic Liquids: Thermal Analysis, Conductivity, and Morphology. Macromolecular Chemistry and Physics 2013, 214, 2099–2107), and minor tendency to develop resistance (Michaud, M.E.; Allen, R.A.; Morrison-Lewis, K.R.; Sanchez, C.A.; Minbiole, K.P.C.; Post, S.J.; Wuest, W.M. Quaternary Phosphonium Compound Unveiled as a Potent Disinfectant against Highly Resistant Acinetobacter Baumannii Clinical Iso-lates. ACS Infectious Diseases 2022, 8, 2307–2314, doi:10.1021/acsinfecdis.2c00382). Additionally, as demonstrated in Section 4 of the present manuscript, generally the synthesis of QPSs is very simple and in most cases involve low-cost reagents, one step reactions, as well as simple purification procedures. As reported in Cieniecka-Rosłonkiewicz, A.; Pernak, J.; Kubis-Feder, J.; Ramani, A.; Robertson, A.J.; Seddon, K.R. Synthesis, An-ti-Microbial Activities and Anti-Electrostatic Properties of Phosphonium-Based Ionic Liquids. Green Chem. 2005, 7, 855–862, doi:10.1039/B508499G, “The reaction conditions are mild, the workup procedure is simple and the yields are high. All of the obtained phosphonium ionic liquids are air- and moisture-stable under ambient conditions”.
On the other hand, to please the Reviewer, some expressions have been toned down along all manuscript and "efficient" has been removed from the title.
Citation line 341:” However, a paucity of innovation in this space has driven the emergence of QACs resistance”. So, I don’t agree with this statement. This statement is not underlined with facts. Moreover, QACs are widely on the market and used extensively for more than 80 years, in many applications without inducing resistance. It would be astonishing if they can now suddenly induce resistance. Moreover, phosphonium are very interesting compound and there is no need to go in competition with arguments which are not justified. The authors mentioned in the conclusions a lot of advantages (line 491) of quaternary phosphonium compounds over quaternary ammonium compounds, which are not discussed in the paper.
I apologize in advance with the Reviewer, but I am forced to note him/her that the signalled sentence is really underlined with facts. The sentence has been found in literature and the related reference was present already in the original version of my manuscript (Ref 168) now Ref. 45 (revised version). However, for more clarity, the reference has been now inserted also at line 394 (revised version). In addition, I recognize that QACs are widely on the market and used extensively for more than 80 years, in many applications, but, perhaps due to their extensive use, they have developed resistance, which is documented, as well. In this regard, please, consider:
Michaud, M.E.; Allen, R.A.; Morrison-Lewis, K.R.; Sanchez, C.A.; Minbiole, K.P.C.; Post, S.J.; Wuest, W.M. Quaternary Phosphonium Compound Unveiled as a Potent Disinfectant against Highly Resistant Acinetobacter Baumannii Clinical Iso-lates. ACS Infectious Diseases 2022, 8, 2307–2314, doi:10.1021/acsinfecdis.2c00382.
Laura, M.T. et al. CS Infect. Dis. 2023, 9, 3, 609–616. Publication Date:February 9, 2023
https://doi.org/10.1021/acsinfecdis.2c00575.
Merchel Piovesan Pereira, B.; Tagkopoulos, I. Benzalkonium Chlorides: Uses, Regulatory Status, and Microbial Resistance. Appl. Environ. Microbiol. 2019, 85, e00377– 19, DOI: 10.1128/AEM.00377-19.
Lee, C. M. et al. Different clinical characteristics and impact of carbapenem-resistance on outcomes between Acinetobacter baumannii and Pseudomonas aeruginosa bacteraemia: a prospective observational study. Sci. Rep. 2022, 12, 8527, DOI: 10.1038/s41598-022-12482-0.
Pogue, J. M.; Zhou, Y.; Kanakamedala, H.; Cai, B. Burden of illness in carbapenem-resistant Acinetobacter baumannii infections in US hospitals between 2014 and 2019. BMC Infect. Dis. 2022, 22, 36, DOI: 10.1186/s12879-021-07024-4.
Bakht, M.; Alizadeh, S. A.; Rahimi, S.; Kazemzadeh Anari, R.; Rostamani, M.; Javadi, A.; Peymani, A.; Marashi, S. M. A.; Nikkhahi, F. Phenotype and genetic determination of resistance to common disinfectants among biofilm-producing and non-producing Pseudomonas aeruginosa strains from clinical specimens in Iran. BMC Microbiol. 2022, 22, 124, DOI: 10.1186/s12866-022-02524-y.
Tong, C.; Hu, H.; Chen, G.; Li, Z.; Li, A.; Zhang, J. Disinfectant resistance in bacteria: Mechanisms, spread, and resolution strategies. Environ. Res. 2021, 195, 110897, DOI: 10.1016/j.envres.2021.110897.
Concerning the advantages of using QPSs reported in the conclusions, the related discussions in the text are present in lines: 324-327, 332-342, 362-270 and 391-406.
The paper is composed of two parts. The first part is a review on bacteria involved by implants. The second part is on quaternary ammonium and phosphonium biocides that can on kill these bacteria. This combination fits quite well.
It is a comprehensive overview and worthwhile to publish. However, it is exaggerated to state that the quaternary phosphonium compounds are preferred over QACs, as they are much more elaborated to prepare.
I am very thankful to the Reviewer for his positive comments on my work. For the rest, considering more carefully Section 4, where the synthetic procedures to obtain the most promising antibacterial and anti BF QPSs developed so far have been reported, a chemist with expertise in organic synthesis would not see any elaborate procedure, but simple low-cost synthesis routinely carried out in common laboratories.
A few minor remarks:
Line 42: “….to treat [3]” should end with a point (.).
The missing point has been added.
Table 2: “…Water (up to 97%)” is of course with respect to the “dry” compounds. Maybe better to say: 0-97%.
Thank you so much for having suggested a correction. Now the correct percentage of water in biofilm has been included in Table 2.
Page 7. The images of figure 1 are taken from the literature and modified a bit. The reference should be given.
The Reviewer is right. Precisely, the image was inspired by an image available online at https://www.zmescience.com/feature-post/natural-sciences/biology-reference/microbiology/what-are-biofilms/ (accessed on 17 December 2023). The reference has been inserted in the Figure 1 caption.
Page 24, line 386: Citatiion”….presence of chloride acid, according to Scheme 2”. Chloride acid should be: hydrogen chloride.
Thank you so much for the suggestion. Chloride acid has been replaced with hydrogen chloride (line 438).
Page 28, line 495: Citation:”….to show researcher how simple and low cost they are…”. The syntheses as described in schemes 3-9 are far from simple and low cost”. I understand that the authors would like to promote the phosphonium compounds and their properties are indeed very good, but please don’t exaggerate to call it cheap. Underline the good properties, as they are present. See also the remark on QACs.
Dear Reviewer, as previously reported, as an Organic chemist dealing daily with the synthesis of organic compounds, I assure you that reactions in Schemes 1, 2, 5, 6, 7 and 8 are very simple, in some cases are one-step procedures and the needed reagents are very cheap. Anyway, also the procedures described in Schemes 3, 4 and 9 are not exceptionally difficult. I agree that several are the remarks on QACs, but also that several reviews already exist reporting them, while QPSs are still not sufficiently considered, studied and reported, thus been necessary further claims, like this review. However, to satisfy the Reviewer, the signalled citation has been removed by conclusions.
Reviewer 2 Report
Comments and Suggestions for Authors
This review deals with very relevant aspects in a broad and multidisciplinary way. But there are several aspects that I think need to be improved.
Table 1 does not correctly describe MDR bacteria. Since some of the key problems, such as Pseudomonas aeruginosa XDR and vancomycin-resistant Enterococcus, are not described in the table.
Figure 1 and 2 are not of sufficient quality.
-Quaternary Phosphonium Salts (QPS). Please structure in paragraphs
- I have detected inappropriate self-citations by authors as reference 5, there are other more appropriate references.
Comments on the Quality of English LanguageI think the English is adequate, although in many cases the text is not well structured, with some paragraphs being too long, making it difficult to read.
Author Response
This review deals with very relevant aspects in a broad and multidisciplinary way. But there are several aspects that I think need to be improved.
Table 1 does not correctly describe MDR bacteria. Since some of the key problems, such as Pseudomonas aeruginosa XDR and vancomycin-resistant Enterococcus, are not described in the table.
As suggested, P. aeruginosa and enterococci have been included in Table 1. In addition, since belonging to ESKAPE bacteria, Acinetobacter baumanni has been included in Table 1, as well.
Figure 1 and 2 are not of sufficient quality.
Figure 1 has been improved. An image with 700 dpi resolution have been provided. Figure 2 has been removed on request of another Reviewer.
-Quaternary Phosphonium Salts (QPS). Please structure in paragraphs
As asked the chemical structures of the antimicrobial QPSs developed in the last years have been inserted in Section 3.3, as Chart.1.
- I have detected inappropriate self-citations by authors as reference 5, there are other more appropriate references.
As suggested, Ref. 5 has been substituted.
Comments on the Quality of English Language
I think the English is adequate, although in many cases the text is not well structured, with some paragraphs being too long, making it difficult to read.
Where necessary, too long paragraphs have been shortened to make the reading of the paper more comfortable.
Reviewer 3 Report
Comments and Suggestions for Authors
Dear Authors:
I read your manuscript with great interest. I believe it will attract lots of readers in the related field. However, to improve your manuscript much better, let me propose for some revisions.
#1: You should set the introduction part where your paper's standpoint in the research area and current trends would be clearer.
#2: Would you add the schematic figures for quaternary ammonium compounds and phosphonium salts to affect bacteria and biofims. And you should describe the difference of mechanisms more in detail.
#3: I think that your tables in this manuscript will be very useful. But the tables relating to QASs and QPSs are missing. Please make them somewhere.
#4: Would you please add the discussion part and summarize the key point of the shifting with mechanism discussion?
Comments on the Quality of English LanguageAs for English, I don't feel any discomfort. But there may be still small errors.
Author Response
Dear Authors:
I read your manuscript with great interest. I believe it will attract lots of readers in the related field. However, to improve your manuscript much better, let me propose for some revisions.
#1: You should set the introduction part where your paper's standpoint in the research area and current trends would be clearer.
As asked, in Section 1 the paper’s standpoints and the current trends in the field have been clarified. Please, see at lines 68-83.
#2: Would you add the schematic figures for quaternary ammonium compounds and phosphonium salts to affect bacteria and biofims. And you should describe the difference of mechanisms more in detail.
As asked, a schematic Figure of the most recognized mode of action of the antimicrobial QASs has been inserted in Section 3.1. as new Figure 2. Since the main mechanism of action of QPSs is like that of QASs, no new scheme has been inserted for these compounds, but some differences in their mode of action respect to QASs have been reported in the main text. Please, see at lines 334-342.
#3: I think that your tables in this manuscript will be very useful. But the tables relating to QASs and QPSs are missing. Please make them somewhere.
Please, see at Table 5 and Table 6. They are the Tables related to QASs and QPSs respectively, not found by the Reviewer.
#4: Would you please add the discussion part and summarize the key point of the shifting with mechanism discussion?
I apologize in advance to the Reviewer, but the meaning of his/her requests is not clear to me. Since in this revision round, I have added further details on the mechanism of action of QASs and QPSs, and the existing small differences between their mechanisms have been now highlighted, I hope to have satisfied the Reviewer's requests in this point by addressing the previous points.
Comments on the Quality of English Language
As for English, I don't feel any discomfort. But there may be still small errors.
All manuscript has been revised with the help of Professor Deirdre Kantz, my colleague English mother tongue and working at the University of Genoa and Pavia (Italy).
Reviewer 4 Report
Comments and Suggestions for Authors
The idea of the review seemed very promising, but the implementation frankly let us down.
Notes:
Table 1 is not relevant to the topic of the study.
Table 2 is meaningless; one sentence in the text is enough.
Table 3 is not relevant to the topic of the study.
Table 3 (No. 2) is meaningless. A couple of sentences in the text would be enough.
Table 3 (No. 2) should be Table 4.
Figure 1. Taken from the article Monroe D. Looking for chinks in the armor of bacterial biofilms. PLoS Biol. 2007 Nov;5(11):e307. doi: 10.1371/journal.pbio.0050307. (slightly amemded)
Figure 2. A good drawing for a popular science magazine
Table 4 is not relevant to the topic of the study.
Figure 4 Wrong or incomplete statistical analysis of publications.
Scheme 1 to 9. Pointless schemes
It's pointless to even discuss the text.
Author Response
The idea of the review seemed very promising, but the implementation frankly let us down.
Notes:
Table 1 is not relevant to the topic of the study.
Why does the Reviewer consider Table 1 not relevant? As reported in the title, abstract, introduction and conclusions of the paper, the main scope of this manuscript was to review QASs and QPSs, the latter being more promising to develop next generation antibiotics, but still too little studied. In particular, my the QASs and QPSs of main interest were those effective in counteracting bacterial resistance and biofilm. In this regard, I have considered necessary to provide readers with an as complete as possible background on bacterial resistance and biofilm, before discussing QASs and QPSs. Table 1 reports the most clinically relevant MDR pathogens, with the associated infections and is essential to the work. In this regard, the Academic Editor himself, far from suggesting removing Table 1, has asked me to add other important pathogens that were missing.
Table 2 is meaningless; one sentence in the text is enough.
One sentence in the text would be not sufficient to provide readers with the information included in Table 2, and the schematic organization of Tables is more readers friendly.
Table 3 is not relevant to the topic of the study.
Why does the Reviewer consider Table 3 not relevant? As reported in the previous point (1), I retained necessary providing readers with an as complete as possible background on bacterial resistance and biofilm, before discussing QASs and QPSs. Table 3, reporting the most clinically relevant BF-producing bacteria and related infections, is essential to the work.
Table 3 (No. 2) is meaningless. A couple of sentences in the text would be enough.
I apologize with the Reviewer for my distraction. Table 4 (and not Table 3) is the correct name of this Table. Concerning the comments of the Reviewer, a couple of sentences in the text would be not sufficient to provide readers with the information included in Table 4, and the schematic organization of Tables is more readers friendly.
Table 3 (No. 2) should be Table 4.
The name of Table has been correct, as well as those of subsequent Tables.
Figure 1. Taken from the article Monroe D. Looking for chinks in the armor of bacterial biofilms. PLoS Biol. 2007 Nov;5(11):e307. doi: 10.1371/journal.pbio.0050307. (slightly amemded)
Precisely, Figure 1 has been created taking inspiration from an image present online at https://www.zmescience.com/feature-post/natural-sciences/biology-reference/microbiology/what-are-biofilms/ (accessed on 17 December 2023), as added in the Figure caption. Probably, this image has been in turn inspired by Figure 1 present in the article cited by the Reviewer. Anyway, such article is an open-article distributed under the terms of the Creative Commons Attribution License, which permits unrestricted use, distribution, and reproduction in any medium.
Figure 2. A good drawing for a popular science magazine
Figure 2 has been removed.
Table 4 is not relevant to the topic of the study.
Table 4 (unrevised manuscript) has been removed.
Figure 4 Wrong or incomplete statistical analysis of publications.
We make kindly note to the Reviewer that data in Figure 4 refers only to experimental works (not review) on ammonium and phosphonium salt reported to be active also against biofilm. I have repeated the survey also crossing data by Scopus with those by PubMed and results are correct. Anyway, a work of year 2022 that was missing in Table 6 has been included, while specification on these questions have been included in the text (lines 379-384 and 386-390).
Scheme 1 to 9. Pointless schemes
It's pointless to even discuss the text.
I apologize with the Reviewer, but it seems that everything in my Review is not relevant or pointless. Rection Schemes providing the synthetic procedures used by chemists to prepare the most relevant antibacterial phosphonium salts which have proved activity also against BF-producing bacteria are essential to stimulate further research on such class of compounds, still not properly considered, and the synthesis of new ones to enlarge the current knowledge and develop a next generation of effective antibiotics.
Round 2
Reviewer 2 Report
Comments and Suggestions for Authors
Now that the authors have reviewed the article, I have no further comments to add.
Author Response
Dear Reviewer,
I am very glad to hear from you the deired poitive comment. Thank you for the work you have done for me.